# Probability Assessment of the Mechanical and Low-Cycle Properties of Structural Steels and Aluminium

**Žilvinas Bazaras, Vaidas Lukoševičius \*, Andrius Vilkauskas and Ramūnas Česnavičius** 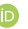

Faculty of Mechanical Engineering and Design, Kaunas University of Technology, Studentų Str. 56, 51424 Kaunas, Lithuania; zilvinas.bazaras@ktu.lt (Ž.B.); andrius.vilkauskas@ktu.lt (A.V.); ramunas.cesnavicius@ktu.lt (R.Č.)
\* Correspondence: vaidas.lukosevicius@ktu.lt

**Abstract:** Key mechanical properties used in low-cycle strength and durability calculations are the strength (proportional limit stress, $\sigma_{pr}$; relative yield strength, $\sigma_{0.2}$; and ultimate tensile stress, $\sigma_u$) and strain properties (proportional limit strain, $e_{pr}$; percent area reduction, $\psi$; and percent area reduction at failure, $\psi_u$). When selecting the key mechanical properties provided in the specifications, an error may be made due to the failure to account for a series of random factors that determine the distribution of properties. The majority of research papers dealing with statistical descriptions of the low-cycle strain properties do not look deeper into the distribution of mechanical properties and the diagram parameters of strain characteristics. This paper provides a description of the distribution patterns of mechanical properties, statistical parameters, and low-cycle fatigue curves. Log-normal distribution generated the lowest values for the coefficient of variation of one of the key statistical indicators, suggesting that log-normal distribution is superior to normal or Weibull distribution in this respect. The distribution of low-cycle strain parameters exceeded the distribution of mechanical properties considerably. Minimum coefficients of variation of the parameters were generated at normal distribution. The statistical analysis showed the lower distribution of the durability parameters compared to the distribution of parameters of the strain diagrams. The findings of the paper enable a revision of the durability and life of the structural elements of in-service facilities subject to elastoplastic loading by assessing the distribution of mechanical characteristics and low-cycle strain parameters as well as the permissible distribution limits.

**Keywords:** probability; durability; low-cycle fatigue; normal distributions; log-normal distribution; Weibull distribution; coefficient of variation

## 1. Introduction

Contemporary transport engineering facilities operate at high speeds, high productivity, and high capacities to achieve the best performance. For aerospace and transport engineering, the equipment and facilities perform under high stress, which may result in elastic–plastic cyclic deformation. Overloading present particular dangers, as cyclically varying loads exceed the proportional limit of the material and cause plastic deformation and the formation of a hysteresis loop. As a result, the durability of the material decreases by hundreds or thousands of cycles.

A wise range of fatigue life prediction methods and probabilistic approaches, as well as mechanical and low cycle properties have been investigated in recent years. A considerable contribution to the calculation of probabilistic methods for mechanical and low-cycle properties was made by a series of investigators. Daunys et al. [1–4] investigated the dependences of the low-cycle durability of mechanical properties for steels of welded joints used in nuclear power plants. Ellingwood et al. [5,6] investigated the applicability of existing statistical data for describing the resistance of steel and reinforced concrete used in nuclear power plants. Liu et al. [7] proposed calculating the equivalent initial flaw size (EIFS) distribution, which is very efficient for calculating the statistics of EIFS.

Xiang et al. [8] proposed a general probabilistic life prediction methodology for accurate and efficient fatigue prognosis, which is based on the inverse first-order reliability method (IFORM) to evaluate the fatigue life at an arbitrary reliability level. Bazaras et al. [9] and Raslavičius et al. [10] investigated the low-cycle durability of nuclear power plants' WWER (Water–Water Energetic Reactor) steels 22k and 15Cr2MoVA. Zhu et al. [11] developed a probabilistic methodology for low-cycle fatigue life prediction using an energy-based damage parameter under Bayes' theorem. Fekete [12] proposed a new low-cycle fatigue prediction model based on strain energy to account for only part of the strain energy stored in the microstructure of the material that causes fatigue damage. Strzelecki [13] proposed the characteristics of the S–N curve using two-parameter and three-parameter Weibull distribution for fatigue limit and limited life. It was demonstrated that S–N curves can be used to determine the fatigue life for a low probability of failure when using a normal distribution. Kosturek et al. [14] presented the results of their research on the low-cycle fatigue properties of Sc-modified AA2519-T62 extrusion. The basic mechanical properties have been established by using tensile tests and low-cycle fatigue testing has been performed on five different levels of total strain amplitude. Manouchehrynia et al. [15] presented a mathematical model to estimate the strain-life probabilistic modelling based on the fatigue reliability prediction of an automobile coil spring under random strain loads. The obtained results demonstrated good agreement between the predicted fatigue lives of the proposed probabilistic model and the measured strain fatigue life models. Lamnauer et al. [16] suggested the use of a probabilistic statistical model for calculating the strength of parts under cyclic fatigue loads. Statistical analysis of the samples (the average value, the corrected variance, the squared asymmetry coefficient, and the excess coefficient) was carried out according to the results of a mass experiment on the strength of samples during fatigue tests. Makhutov et al. [17] analysed traditional engineering methods for the assessment of the lifetime characteristics of fatigue resistance. The methods used were based on deterministic parameters. The authors presented the results of experimental studies and the calculations of strength and durability for low-alloy and austenitic steels with varying mechanical properties.

Durability is one of the key criteria of structural elements. The application of appropriate probability calculation methods is important in the pursuit of extended life for in-service facilities. They also contribute to more accurate and research-based determinations of the safety values of cyclic loads at the design phase. Low-cycle strength and durability calculations based on the guaranteed mechanical characteristics rather than the standard ones retrieved from the specifications are necessary for the determination of the strength safety margin of structural elements. The strength safety margin of structural elements is, in turn, necessary for the assessment of the reliability of operation of the critical structures [18–20].

The majority of the studies that undertake statistical assessment of low-cycle fatigue are focused on the uniaxial strain state and assessment of the durability distribution until the initiation of the fatigue crack or until the crack reaches a certain length. Currently, there are no consistent studies on the construction of probability curves for low-cycle fatigue in view of the values of the guaranteed mechanical characteristics [21–26].

Based on the topics discussed above, the main contributions of this paper are as follows: (1) We determine the distribution patterns of the mechanical properties, statistical parameters, and low-cycle fatigue curves; (2) we perform an analysis of the statistical assessment of cyclic elastoplastic strain diagrams and of the parameters; (3) we refine the low-cycle strength and durability calculations based on the verified values rather than standard values of key mechanical properties; and (4) we present a comparison of the low-cycle fatigue probability curves of the experimental data.

## 2. Materials and Methods

Experiments were conducted at the Laboratory of the Mechanical Engineering Department at Kaunas Technology University. The experimental equipment used for fatigue tests

consisted of a 50 kN testing machine Instron (Norwood, MA, USA) 8801 series Servo Hydraulic Fatigue testing system with a FastTrack (Norwood, MA, USA) 8800 controller and an electronic device that was designed to record the stress–strain curves. A deformation rate of four cycles per minute was used for the fatigue tests. The mechanical characteristics were measured with an error not exceeding ±1% of the deformation scale. For the dynamic inertia compensation during fatigue testing, the Dynacell (Norwood, MA, USA) Dynamic Load Cell ±250 N was used (corresponding to ISO 75001/1 Class 0.5, ISO 10002 Part 2, ASTM E4, EN10002 Part 2, and JIS (B7721, B7733).

Fatigue tests have been performed in accordance with the GOST 25502-79 standard (Strength analysis and testing in machine building; Methods of metals mechanical testing; Methods of fatigue testing) [27]. Standard GOST 22015-76 (Quality of product; Regulation and statistical quality evaluation of metal materials and products on speed-torque characteristics) [28] was used to calculate the statistical characteristics.

The main task of the fatigue loading device is to give the deformable specimen a homogeneous state of deformation. The specimens used for the cyclic deformation tests under the linear stress state ensure a homogeneous stress state in the test piece until a fatigue crack occurs. A drawing of the sample best meeting these conditions is given in Figure 1a.

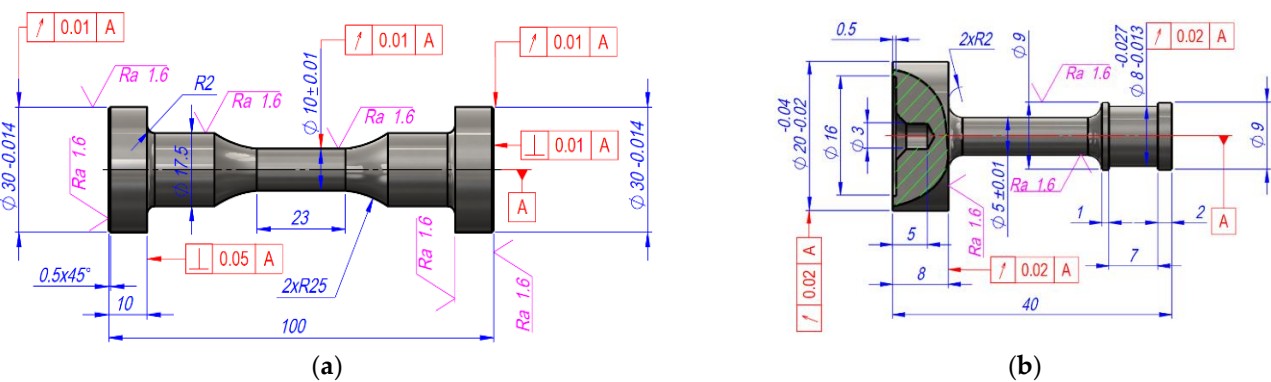

(**a**)  (**b**)

**Figure 1.** Shape and dimensions of the specimens for: (**a**) low-cycle fatigue tension-compression experiments; (**b**) monotonous tensile experiments.

After the fatigue tests, the fractured specimens were used as the workpiece materials to produce monotonous tensile specimens, with the aim of creating material properties nearly identical to those of the material subjected to cyclic loading. The monotonous tensile specimens of a circular cross section with d = 9 mm and l = 40 were taken from the parts of cyclic test specimens, which had not been subjected to plastic deformation (Figure 1b). Tables 1 and 2 show the mechanical properties and chemical composition of cyclically softening (alloyed steel 15Cr2MoVA), stable (structural steel C45), and hardening (aluminium alloy D16T1) materials.

The mechanical characteristics required for the investigation were generated by experiments on the specimens made of three materials: alloyed steel 15Cr2MoVA (160 specimens), structural steel 45 (220 specimens), and aluminium alloy D16T1 (120 specimens). Different numbers of samples were used to determine the effect of the sample size on the statistical results. All test equipment and methods are described in detail in the literature [29].

**Table 1.** The chemical composition of the materials.

| Material | C | Si | Mn | Cr | Ni | Mo | V | S | P | Mg | Cu | Al |
|---|---|---|---|---|---|---|---|---|---|---|---|---|
|  |  |  |  |  |  | % |  |  |  |  |  |  |
| 15Cr2MoVA (GOST 5632-2014) | 0.18 | 0.27 | 0.43 | 2.7 | 0.17 | 0.67 | 0.30 | 0.019 | 0.013 | - | - | - |
| C45 (GOST 1050-2013) | 0.46 | 0.28 | 0.63 | 0.18 | 0.22 | - | - | 0.038 | 0.035 | - | - | - |
| D16T1 (GOST 4784-97) | - | - | 0.70 | - | - | - | - | - | - | 1.6 | 4.5 | 9.32 |

**Table 2.** Mechanical properties of the materials.

| Material | $e_{pr}$ | $\sigma_{pr}$ | $\sigma_{0.2}$ | $\sigma_u$ | $S_k$ | $\psi$ |
|---|---|---|---|---|---|---|
| | % | MPa | | | | % |
| 15Cr2MoVA (GOST 5632-2014) | 0.200 | 280 | 400 | 580 | 1560 | 80 |
| C45 (GOST 1050-2013) | 0.260 | 340 | 340 | 800 | 1150 | 39 |
| D16T1 (GOST 4784-97) | 0.600 | 290 | 350 | 680 | 780 | 14 |

## 3. Identification of the Key Mechanical Properties and Correlations between Them

The standard (reference) key mechanical properties of materials were used as the input data for the calculation of the distribution under the probability methods for low-cycle strength and durability calculations [30–32]. Hence, it was necessary to determine the preferable theoretical laws applicable to the experimental distribution functions of the key mechanical properties. The relationships between the mechanical properties simultaneously had to be determined in order to substantiate their values at a certain probability level.

Due to the very large number of results (on the order of hundreds) generated by the multiple tests, additional statistical processing—namely, statistical data series—was performed for the identification of the mechanical properties of the materials. For the histograms, the total array of the results was divided into 10 equal bins (statistical intervals). Their width was calculated using the following equation:

$$x_{\text{int}} = \frac{x_{\max} - x_{\min}}{n_{\text{int}} - 1}.$$ (1)

Exceeding 15–20 intervals would have been unreasonable due to the fact that even a very large number of results might still not ensure the accuracy of statistical characteristics. Following the division of the statistical data series into 10 intervals, the following histograms were developed (Figures 2–4).

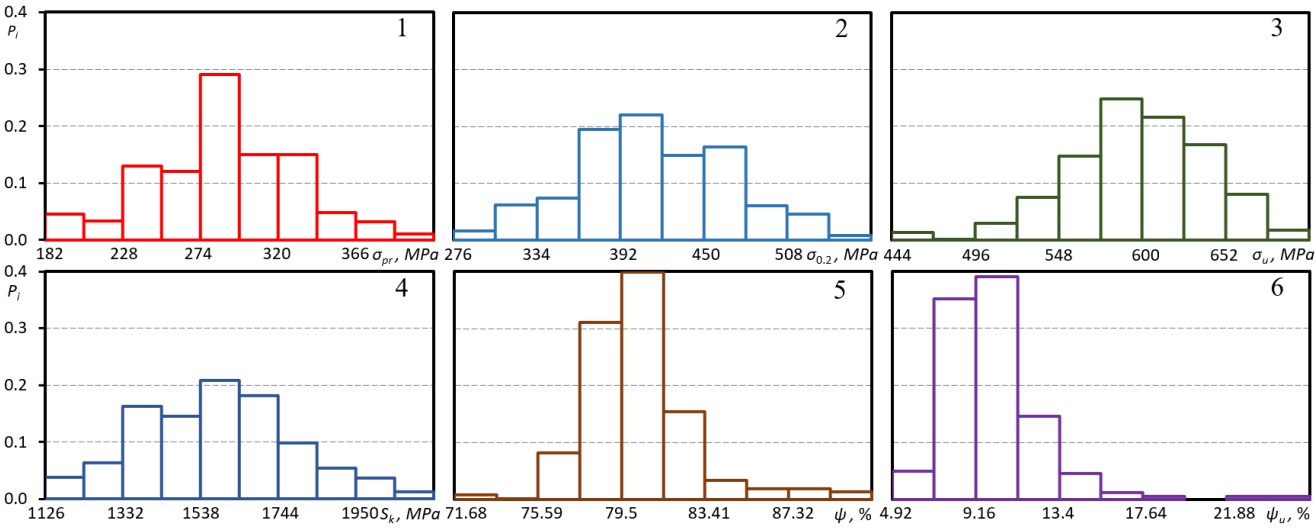

**Figure 2.** Histograms of mechanical properties of steel 15Cr2MoVa (1—$\sigma_{pr}$; 2—$\sigma_{0.2}$; 3—$\sigma_u$; 4—$S_k$; 5—$\psi$; and 6—$\psi_u$).

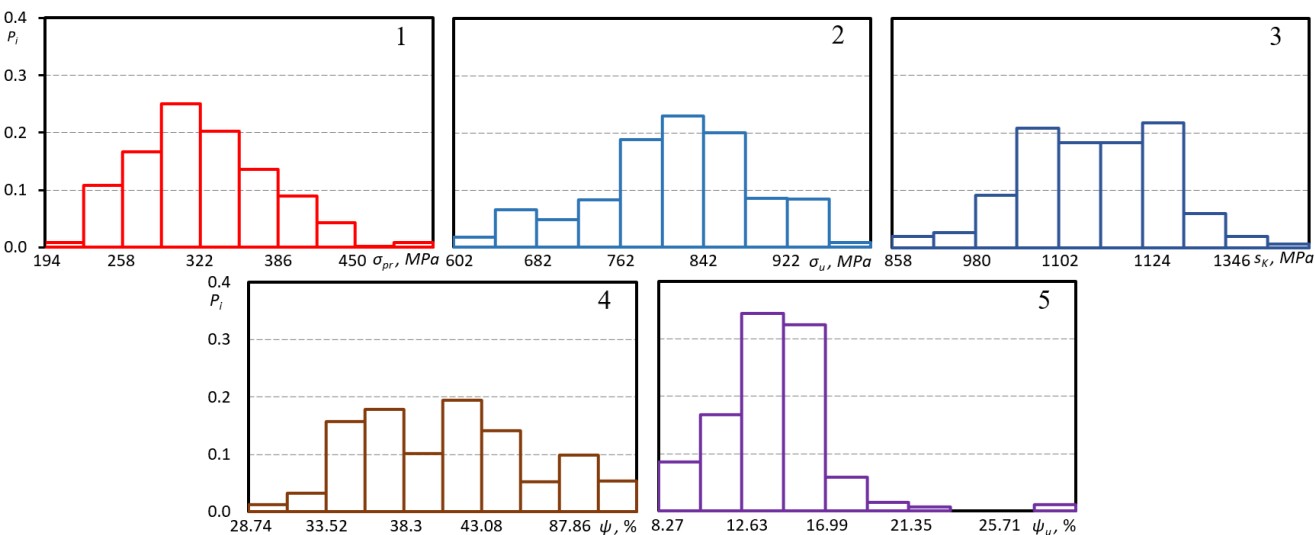

**Figure 3.** Histograms of mechanical properties of steel C45 (1—$\sigma_{pr}$; 2—$\sigma_u$; 3—$S_k$; 4—$\psi$; and 5—$\psi_u$).

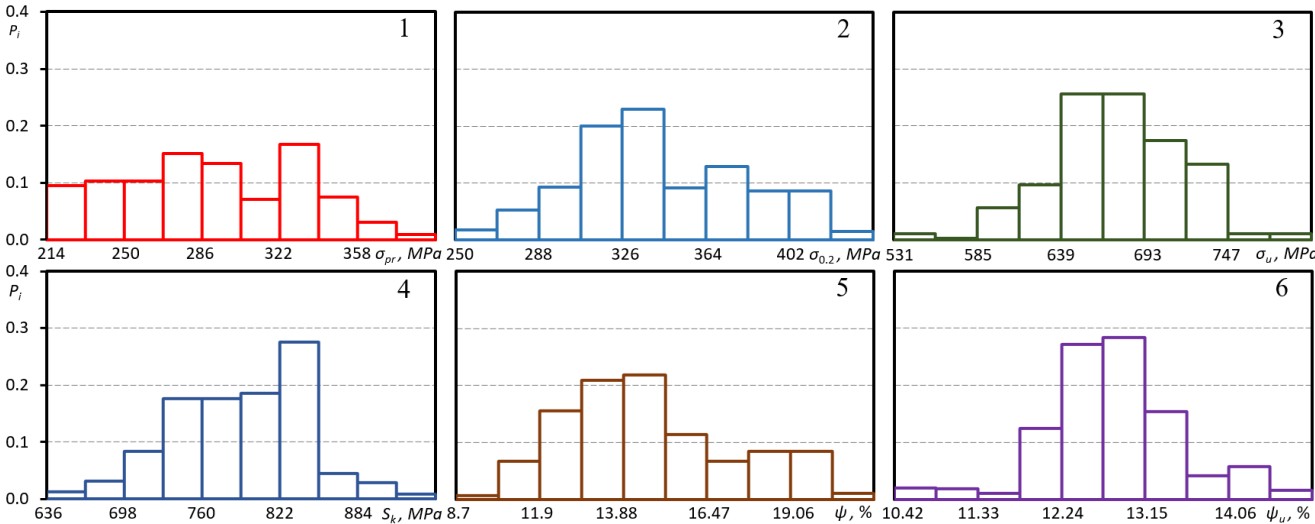

**Figure 4.** Histograms of mechanical properties of aluminium alloy D16T1 (1—$\sigma_{pr}$; 2—$\sigma_{0.2}$; 3—$\sigma_u$; 4—$S_k$; 5—$\psi$; and 6—$\psi_u$).

Intervals equal in length were marked on the abscissa axis and the height of each interval was calculated using the following equation:

$$P_i = \frac{m_i}{n}. \tag{2}$$

The histogram analysis showed the qualitative correspondence of the mechanical properties to the normal distribution law. Nonetheless, in order to improve the statistical assessment of the properties, statistical characteristics were calculated under the three applicable distribution laws: normal, log-normal, and Weibull distribution. The key statistical characteristics were calculated for the normal and log-normal distribution:

$$\overline{x} = \frac{1}{n}\sum_{i=1}^{n} x_i, \; s = \sqrt{\frac{1}{n-1}\sum_{i=1}^{n}(x_i - \overline{x})^2}, \; D = s^2, \; S = \frac{m^3}{s^3} = \frac{\sum_{i=1}^{n}(x_i - \overline{x})^3}{s^3 \times n}, \; V = \frac{s}{\overline{x}}. \tag{3}$$

For the Weibull distribution [33]:

$$\bar{x} = bk_b + c, \; s = bk_b, \; S = \frac{\frac{n}{(n-1)(n-2)}\sum_{i=1}^{n}(x_i-\bar{x})^3}{\left[\left(\frac{1}{n-1}\right)\sum_{i=1}^{n}x_i-\bar{x}\right]^{3/2}}, \; V = \frac{bk_b}{bk_b+c}, \; b = \frac{s}{q_b}. \tag{4}$$

Table 3 shows the calculated key statistical normal, log-normal, and Weibull distribution characteristics of the mechanical properties of the materials investigated.

**Table 3.** Statistical characteristics for normal, log-normal, and Weibull distribution.

| Mechanical Property | Material | Normal | | | | | Log-Normal | | | | | Weibull | | | |
|---|---|---|---|---|---|---|---|---|---|---|---|---|---|---|---|
| | | $\bar{x}$ | s | D | S | V | $\bar{x}$ | s | D | S | V | $\bar{x}$ | s | S | V |
| $\sigma_{pr}$, MPa | 15Cr2MoVa | 287 | 42 | 1764 | 0.080 | 0.146 | 284 | 0.0649 | 0.0042 | −0.301 | 0.026 | 287 | 42 | 0.081 | 0.146 |
| | 45 | 325 | 52 | 2704 | 0.410 | 0.160 | 321 | 0.0688 | 0.0047 | 0.074 | 0.027 | 325 | 52 | 0.419 | 0.159 |
| | D16T1 | 292 | 40 | 1600 | −0.043 | 0.136 | 290 | 0.0609 | 0.0037 | −0.263 | 0.025 | 292 | 40 | −0.043 | 0.138 |
| $\sigma_{0.2}$, MPa | 15Cr2MoVa | 414 | 53 | 2809 | 0.002 | 0.128 | 410 | 0.0571 | 0.0032 | −0.299 | 0.022 | 414 | 53 | 0.002 | 0.129 |
| | 45 | 325 | 52 | 2704 | 0.410 | 0.160 | 321 | 0.0688 | 0.0047 | 0.073 | 0.027 | 325 | 52 | 0.418 | 0.159 |
| | D16T1 | 346 | 41 | 1681 | 0.011 | 0.118 | 343 | 0.0513 | 0.0026 | 0.117 | 0.020 | 346 | 41 | 0.023 | 0.117 |
| $\sigma_u$, MPa | 15Cr2MoVa | 602 | 42 | 1764 | −0.491 | 0.069 | 600 | 0.0315 | 0.0009 | −0.828 | 0.011 | 602 | 42 | −0.499 | 0.070 |
| | 45 | 811 | 77 | 5929 | −0.033 | 0.095 | 806 | 0.0423 | 0.0018 | −0.558 | 0.015 | 811 | 77 | −0.340 | 0.095 |
| | D16T1 | 677 | 41 | 1681 | −0.141 | 0.061 | 676 | 0.0267 | 0.0007 | −0.308 | 0.009 | 677 | 41 | −0.142 | 0.061 |
| $S_k$, MPa | 15Cr2MoVa | 1585 | 201 | 40,401 | 0.240 | 0.127 | 1571 | 0.0551 | 0.0030 | −0.033 | 0.017 | 1585 | 201 | 0.246 | 0.127 |
| | 45 | 1154 | 100 | 10,000 | −0.110 | 0.087 | 1149 | 0.0381 | 0.0014 | −0.278 | 0.012 | 1154 | 100 | −0.112 | 0.087 |
| | D16T1 | 793 | 52 | 2704 | −0.061 | 0.066 | 791 | 0.0287 | 0.0008 | −0.218 | 0.010 | 793 | 52 | −0.062 | 0.066 |
| $\psi$, % | 15Cr2MoVa | 80.12 | 2.41 | 5.86 | 1.451 | 0.030 | 80.03 | 0.0129 | 0.0002 | 1.186 | 0.007 | 79.60 | 2.42 | 1.478 | 0.032 |
| | 45 | 41.05 | 5.18 | 26.86 | 0.225 | 0.126 | 40.68 | 0.0548 | 0.0029 | 0.045 | 0.034 | 41.05 | 5.18 | 0.228 | 0.126 |
| | D16T1 | 14.59 | 2.58 | 6.64 | 0.529 | 0.177 | 14.36 | 0.0754 | 0.0057 | 0.168 | 0.065 | 14.59 | 2.58 | 0.542 | 0.177 |
| $\psi_u$, % | 15Cr2MoVa | 10.16 | 5.56 | 6.58 | 2.432 | 0.252 | 9.90 | 0.0945 | 0.0089 | 0.995 | 0.095 | 8.19 | 2.56 | 0.248 | 0.313 |
| | 45 | 14.18 | 2.77 | 7.67 | 1.446 | 0.195 | 13.92 | 0.0807 | 0.0065 | 0.258 | 0.071 | 12.75 | 2.77 | 1.466 | 0.217 |
| | D16T1 | 12.84 | 0.71 | 0.51 | 0.071 | 0.055 | 12.82 | 0.0241 | 0.0006 | −0.168 | 0.022 | 12.80 | 0.71 | 0.073 | 0.056 |

Table 3 and the histograms (Figures 2–4) suggest the presence of a fairly large asymmetry of the majority of mechanical properties. The percent area reduction $\psi$ and percent area reduction at failure $\psi_u$ show the largest asymmetry. The coefficient of variation $V$ is one of the key statistical indicators. Its lowest values were obtained using the log-normal distribution. Hence, this distribution may be considered superior to normal or Weibull distribution. It should be noted that the values of both the coefficient of variation and other mechanical properties were similar in the Weibull and normal distribution. The lowest values obtained were those of the ultimate tensile stress $\sigma_u$ and cyclic stress $S_k$ of $k$ semicycle. The abscissa axis of Figure 5 presents the coefficient of variation of Weibull and normal distribution laws and the ordinate axis—the coefficient of variation of log—normal distribution law.

The reliable quantitative assessment of mechanical properties is possible with a large sample available. In the case of a limited number of tests, the degree of accuracy and reliability must be provided, i.e., the confidence intervals, must be calculated (Table 4):

$$\bar{x} - \frac{S}{\sqrt{n-1}}t_{\gamma 1} \leq \mu \leq \bar{x} + \frac{S}{\sqrt{n-1}}t_{\gamma 1}. \tag{5}$$

For higher reliability of the probability calculations of the strength and durability of structural elements, it would be reasonable to use the calculated limit values of confidence intervals rather than the standard mechanical properties provided in the specifications [34].

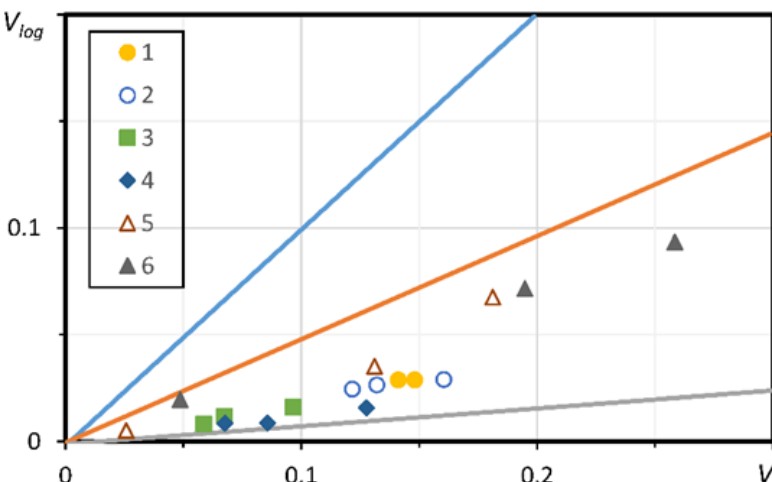

**Figure 5.** Coefficients of variation of mechanical properties (x-axis—normal and Weibull; y-axis—log-normal): 1—$\sigma_{pr}$; 2—$\sigma_{0.2}$; 3—$\sigma_u$; 4—$S_k$; 5—$\psi$; and 6—$\psi_u$.

**Table 4.** The range of confidence intervals of mechanical properties.

| Material Property | $p$ | 15Cr2MoVa | | C45 | | D16T1 | |
|---|---|---|---|---|---|---|---|
| | | $x_p^U$ | $x_p^L$ | $x_p^U$ | $x_p^L$ | $x_p^U$ | $x_p^L$ |
| $\sigma_{pr}$, MPa | 0.01 | 277.9 | 277.4 | 313.4 | 312.8 | 284.5 | 283.9 |
| | 0.50 | 284.4 | 283.8 | 321.4 | 320.8 | 290.1 | 289.6 |
| | 0.99 | 290.9 | 290.3 | 329.7 | 329.1 | 296.1 | 295.5 |
| $\sigma_{0.2}$, MPa | 0.01 | 403.4 | 402.7 | 313.4 | 312.8 | 338.9 | 338.4 |
| | 0.50 | 410.5 | 409.8 | 321.4 | 320.8 | 343.7 | 343.3 |
| | 0.99 | 417.7 | 417.0 | 329.7 | 329.1 | 348.6 | 348.1 |
| $\sigma_u$, MPa | 0.01 | 596.7 | 596.4 | 798.3 | 797.7 | 673.3 | 673.0 |
| | 0.50 | 600.4 | 599.6 | 806.2 | 805.4 | 676.2 | 675.6 |
| | 0.99 | 603.1 | 602.7 | 813.7 | 813.1 | 678.4 | 678.2 |
| $S_k$, MPa | 0.01 | 1546.9 | 1544.8 | 1140.2 | 1139.5 | 787.7 | 787.4 |
| | 0.50 | 1572.3 | 1570.1 | 1149.1 | 1148.4 | 791.2 | 790.8 |
| | 0.99 | 1598.0 | 1595.8 | 1158.8 | 1157.4 | 794.7 | 794.4 |
| $\psi$, % | 0.01 | 79.96 | 79.94 | 40.05 | 40.01 | 13.95 | 13.91 |
| | 0.50 | 80.04 | 80.02 | 40.70 | 40.66 | 14.38 | 14.34 |
| | 0.99 | 80.10 | 80.08 | 41.36 | 41.32 | 14.83 | 14.78 |
| $\psi_u$, % | 0.01 | 9.45 | 9.42 | 13.46 | 13.42 | 12.78 | 12.76 |
| | 0.50 | 9.92 | 9.87 | 13.94 | 13.90 | 12.83 | 12.81 |
| | 0.99 | 10.41 | 10.36 | 14.44 | 14.39 | 12.87 | 12.85 |

It is recommended to replace the mean value of the mechanical property with the lower endpoint of the confidence interval and the standard deviation of mechanical property with the upper endpoint of the confidence interval. Normally, when designing the facilities, standard values of mechanical properties are used. However, irrespective of the existing distribution of the properties, a considerable deviation from the true structural strength and durability is likely. In order to identify the error of the available statistical series, normalised values of mechanical properties were determined under the standard methodology and then compared to the standard values and arithmetic means of the materials under investigation.

In order to ensure a reliable safety margin, the mechanical properties were normalised from the bottom and the calculated norm values were determined using the variation

series in ascending order by the variable. In this case, the calculated norm values were determined using the following equation:

$$c_0 = \bar{x} - k_1 S. \tag{6}$$

The values of the statistical data density quantile $k_1$ depended on the sample size and materials investigated, namely, steel 15Cr2MoVa—1.41; steel C45—1.40; and aluminium alloy D16T1—1.43 [28].

See Table 5 for the calculations of the normalised mechanical properties performed at the required confidence level and under the procedure described above.

**Table 5.** Normalised mechanical properties.

| Mechanical Properties | $c_0$ | | |
|---|---|---|---|
| | **15Cr2MoVa** | **C45** | **D16T1** |
| $\sigma_{pr}$, MPa | 228 | 253 | 235 |
| $\sigma_{0.2}$, MPa | 339 | 253 | 288 |
| $\sigma_u$, MPa | 542 | 703 | 618 |
| $S_k$, MPa | 1302 | 1014 | 719 |
| $\psi$, % | 76.72 | 33.79 | 10.90 |
| $\psi_u$, % | 6.58 | 10.30 | 11.82 |

Along with the normalised mechanical properties, Figure 6 presents the experimental and standard data of log-normal distribution of all the materials investigated. Figure 6 suggests that the results of all the experimental mechanical properties are distributed linearly and this confirms the correspondence of the values to the log-normal distribution. Standard properties are mostly used when calculating the strain and durability diagram parameters: relative yield strength stress, $\sigma_{0.2}$; ultimate tensile stress, $\sigma_u$; and relative percent area reduction, $\psi$. These properties are usually provided in the reference sources.

A comparison of the experimental data, standard, and normalised properties ($\sigma_{0.2}$, $\sigma_u$, and $\psi$) is given in Figure 6, which suggests that the values of the standard properties do not correspond to the experimental and normalised data. High probability is characteristic of the reference properties of the relative yield strength ($\sigma_{0.2}$) of the materials investigated: steel 15Cr2MoVa—74%; steel C45—62%; and aluminium alloy D16T1—90%. Meanwhile, the values of probability of the normalised mechanical properties $\sigma_{0.2}$ are considerably lower: steel 15Cr2MoVa—12%; steel C45—7.5%; and aluminium alloy D16T1—8%. Similar results were obtained for the stress ($\sigma_u$) indicated in the ultimate strength standards for steel 15Cr2MoVa. The values of reference stress $\sigma_u$ of steel C45 and aluminium alloy D16T1 corresponded to a probability lower than 1%. Normalised $\sigma_u$ mechanical properties were: steel 15Cr2MoVa-25%; steel C45—105%; and aluminium alloy D16T1—8%.

The reference value of the relative percent area reduction $\psi$ is 0.0003% for steel 15Cr2MoVa, while for steel C45, it corresponds to the experimental data, i.e., 50%. The normalised value for steel 15Cr2MoVa is 15%, for steel C45 it is 16%, and for aluminium alloy D16T1 it is 5%.

The key property defining the low-cycle durability is the relative percent area reduction $\psi$. Application of the comparative value $\psi$ = 50% with 0.0003% probability for steel 15Cr2MoVa resulted in a very high durability safety factor compared to the standard safety factor. When the normalised value $\psi$ = 76.72% with 15% probability was applied, the resulting deviation of the durability safety factor was fairly small.

For steel C45, a standard $\psi$ value of 40–45%, corresponding to 50% probability, did not provide a sufficient confidence level. In this case, the normalised value $\psi$ = 33.79% corresponded to a 10% probability.

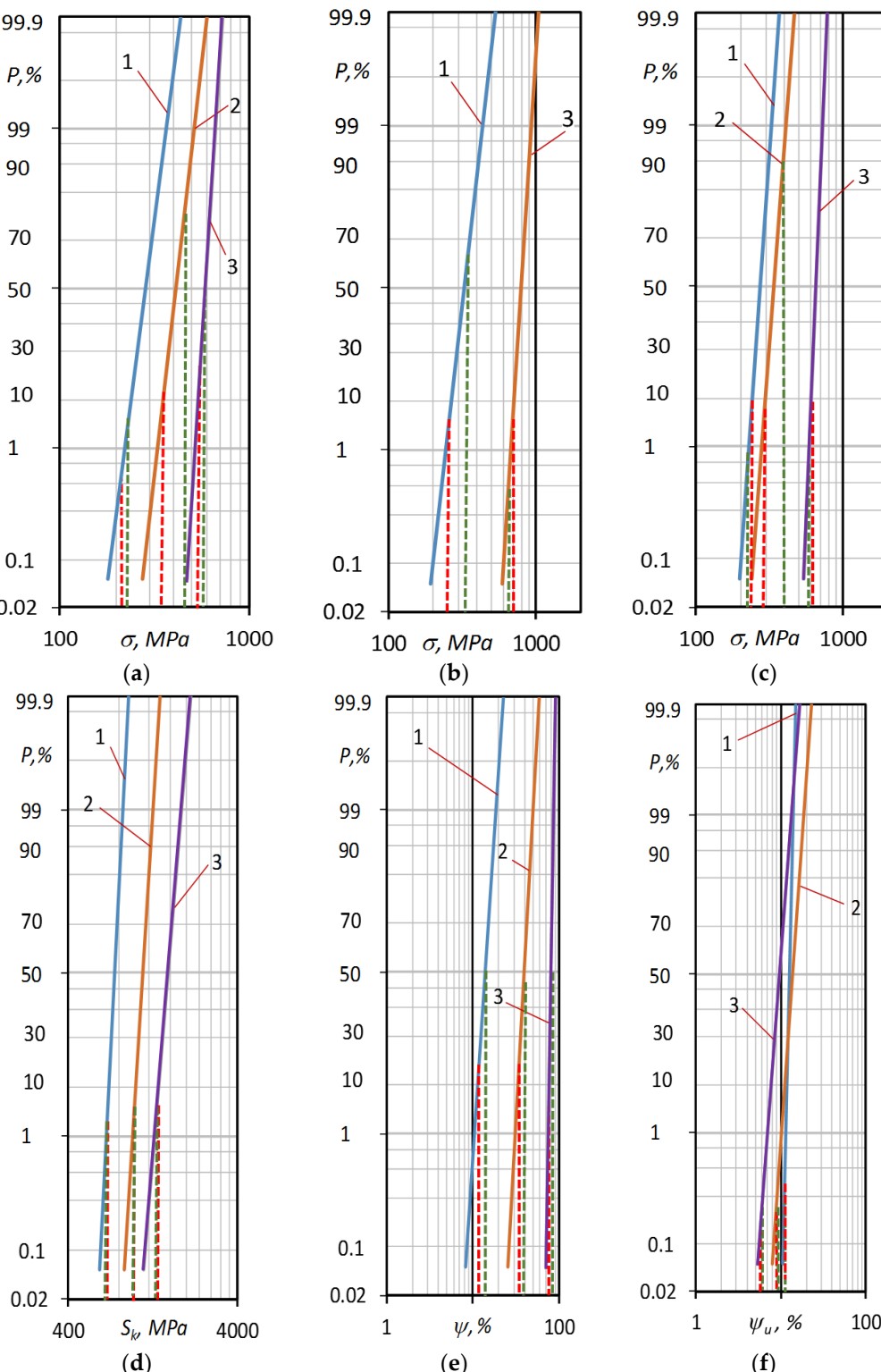

**Figure 6.** Log-normal distribution curves of the proportionality, relative yield, and strength limits: (**a**) steel 15Cr2MoVa; (**b**) steel C45; (**c**) aluminium alloy D16T1 (1—$\sigma_{pr}$; 2—$\sigma 0.2$; and 3—$\sigma_u$); of the stress at failure (**d**), continuous (**e**), and percent area reduction at failure (**f**), 1—steel 15Cr2MoVa; 2—steel C45; 3—aluminium alloy D16T1; ▬ ▬ ▬ —normalised values; ▬ ▬ ▬ —standard values.

The analysis of standard, normalised, and experimental results showed that the application of standard properties to the low-cycle fatigue calculations may lead to significant deviations from the actual results.

The distribution of a random event under the normal distribution law is known to be characterised by mean square deviation $s$ and dispersion $D$. As the mean square deviation $s$ may be mathematically associated with the maximum and minimum values of the random measure, the ratio of the values may also be considered as the distribution characteristic:

$$K = \frac{x_{\max}}{x_{\min}}. \tag{7}$$

Table 6 presents the $K$ values of measures $\sigma_{pr}$, $\sigma_{0.2}$, $\sigma_u$, $S_k$, $\psi$, and $\psi_u$ of the key mechanical properties.

**Table 6.** $K$ values of relative measures of the key mechanical properties.

| Material | $\sigma_{pr}$ | $\sigma_{0.2}$ | $\sigma_u$ | $S_k$ | $\psi$ | $\psi_u$ |
|---|---|---|---|---|---|---|
| | **MPA** | | | | **%** | |
| 15Cr2MoVa | 2.06 | 1.88 | 1.52 | 1.78 | 1.24 | 4.19 |
| C45 | 2.41 | 2.41 | 1.57 | 1.62 | 1.71 | 3.09 |
| D16T1 | 1.72 | 1.66 | 1.45 | 1.43 | 2.24 | 1.38 |

According to Table 6, high values of relative percent area reduction $\psi_u$ were seen for steel 15Cr2MoVa and C45. A comparison of the values of mechanical properties $\sigma_{pr}$ and $\sigma_u$ of the same materials showed that, for steel 15Cr2MoVa, the value of the proportionality limits $\sigma_{pr}$ of the $K$ coefficient was 26% higher than the ultimate strength $\sigma_u$ value. This was 35% for steel C45 and 16% for aluminium alloy D16T1. This was related to the considerable distribution of the properties of the proportionality limit. The diagrams of $K$–$s$ and $K$–$V$ in Figure 7 show the curves of the materials and mechanical properties investigated, described by the following equations [27].

$$s^2 = 0.035(K - 1), \; V^2 = 0.0017(K - 1) \tag{8}$$

The resulting values enabled a primary assessment of the statistical properties $s$, $V$, and $\bar{x}$ according to the marginal values of mechanical properties usually provided in the material specifications. The resulting initial statistical characteristics may also be used for the determination of the minimum number of statistical specimens:

$$n_a = \frac{V^2}{\Delta_a^2} t_{1-\gamma/2}^2. \tag{9}$$

The analysis of calculation results of $n_a$ values (Table 7) showed that the error $\Delta_a$ of determination of the mean value of a random measure had a considerable effect on the number of statistical tests.

A comparison of the calculated number of specimens with the values of the error $\Delta_a$ of determination of the mean value of a random measure (equal to 0.01–0.05) showed that the increase in $\Delta_a$ to 0.05 led to a 10-fold to 30-fold reduction in the number of specimens. In the same manner, the number of specimens was also affected by the reliability of normal distribution $\gamma$. The increase in its value from 0.05 to 0.1 led to a 1.5-fold increase in the number of specimens.

Single strain diagrams had a considerable effect on the low-cycle fatigue tests of the same material. The distribution of the single strain diagrams had a direct effect on the cyclic test diagrams and specimen durability [1]. In Figure 8 the absolute $\sigma$–$e$ and relative $\bar{\sigma}$–$\bar{e}$ coordinates present the single strain diagrams of the limit values of the statistical series of the materials investigated.

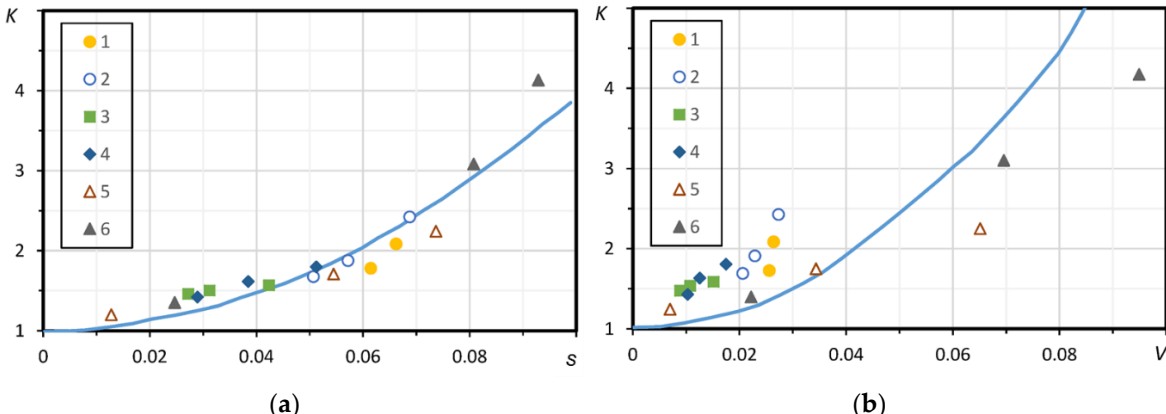

**Figure 7.** Dependence of relative value $K$ under the log-normal distribution law on mean square deviation $s$ (**a**) and coefficient of variation $V$ (**b**) (1—$\sigma_{pr}$; 2—$\sigma_{0.2}$; 3—$\sigma_u$; 4—$S_k$; 5—$\psi$; and 6—$\psi_u$).

**Table 7.** Dependence of the number of specimens on the initial statistical properties.

| Mechanical Characteristic | Material | $\Delta_a$ | | | | | |
|---|---|---|---|---|---|---|---|
| | | 0.01 | | 0.03 | | 0.05 | |
| | | $\gamma$ | | | | | |
| | | 0.05 | 0.10 | 0.05 | 0.10 | 0.05 | 0.10 |
| $\sigma_{pr}$, MPa | 15Cr2MoVa | 819 | 573 | 47 | 33 | 33 | 23 |
| | C45 | 984 | 689 | 110 | 77 | 39 | 28 |
| | D16T1 | 709 | 497 | 78 | 55 | 28 | 20 |
| $\sigma_{0.2}$, MPa | 15Cr2MoVa | 629 | 440 | 71 | 49 | 25 | 18 |
| | C45 | 984 | 689 | 110 | 77 | 39 | 28 |
| | D16T1 | 535 | 375 | 59 | 41 | 21 | 15 |
| $\sigma_u$, MPa | 15Cr2MoVa | 182 | 128 | 20 | 14 | 7 | 5 |
| | C45 | 347 | 243 | 39 | 27 | 14 | 10 |
| | D16T1 | 139 | 97 | 16 | 11 | 6 | 4 |
| $S_k$, MPa | 15Cr2MoVa | 619 | 434 | 69 | 48 | 25 | 17 |
| | C45 | 290 | 203 | 31 | 22 | 12 | 8 |
| | D16T1 | 167 | 117 | 18 | 12 | 7 | 5 |
| $\psi$, % | 15Cr2MoVa | 35 | 25 | 4 | 3 | 1 | 1 |
| | C45 | 609 | 427 | 69 | 48 | 24 | 17 |
| | D16T1 | 1203 | 842 | 133 | 93 | 48 | 34 |
| $\psi_u$, % | 15Cr2MoVa | 2440 | 1708 | 18,286 | 200 | 98 | 68 |
| | C45 | 1460 | 1022 | 162 | 114 | 58 | 41 |
| | D16T1 | 116 | 81 | 14 | 10 | 5 | 3 |

An investigation of the tension diagrams suggested that the distribution of low-cycle test results had been affected by the type of loading and by the absolute or relative loading coordinates used.

In the case of loading with a controlled strain ($e_k$ = constant), the effect was minor and depended on the level of loading. It could be observed in Figure 8 that loading with controlled stress ($S_k$ = constant) was difficult to implement on the absolute coordinates. For steel 15Cr2MoVa, with the load being up to 400 MPa, the strain $e$ varied from 0.2% to 0.4%. Moreover, strain $e$ varied from 0.2% to 4.5%, where loading reached 450 MPa and strain $e$ varied from 0.2% to 11.5%. Where the loading level reached 500 MPa, strain $e$ varied from 0.2% to ∞.

The distribution of strain values on the relative coordinates decreased considerably. According to Figure 8, for steel 15Cr2MoVa the value of relative strain $\bar{e}$ varied from 1 to 3

where $\overline{\sigma} = 1.1$. Meanwhile, where $\overline{\sigma} = 1.4$ the value of relative strain $\overline{e}$ ranged from 3.5% to 55%. Similar results were obtained when analysing the diagrams for steel C45 and aluminium alloy D16T1. For steel C45 (Figure 8c), with the load being up to 300 MPa, strain $e$ varied from 0.2% to 6.5%. Where the load reached 500 MPa, strain $e$ varied from 0.2% to 8.5%, while when $\sigma = 650$ MPa strain $e = 11.5-\infty$. Where the relative load was 1.1, strain $\overline{e}$ varied from 24% to 42% and where $\overline{\sigma} = 1.4$ strain $\overline{e} = 37-70\%$ (Figure 8d). Similar results were obtained for the aluminium alloy D16T1 (Figure 8e,f).

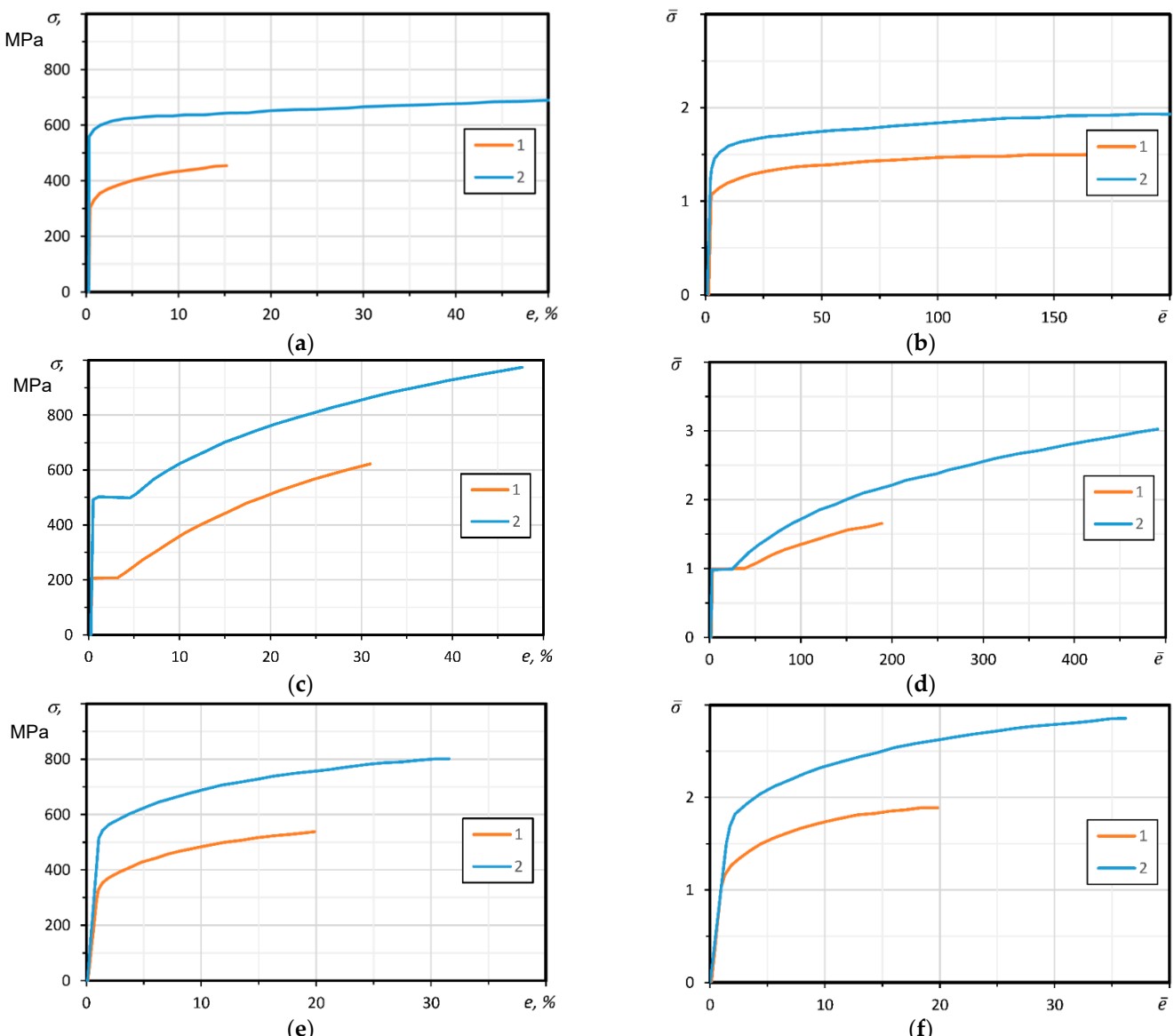

**Figure 8.** Strain diagrams. For steel 15Cr2MoVa: (**a**) on the absolute coordinates $\sigma-e$; (**b**) on the relative coordinates $\overline{\sigma}-\overline{e}$ ($1 - P = 0.3125\%$; $2 - P = 99.6875\%$); steel C45: (**c**) on the absolute coordinates $\sigma-e$; (**d**) on the relative coordinates $\overline{\sigma}-\overline{e}$ ($1 - P = 0.23\%$; $2 - P = 99.77\%$); for aluminium alloy D16TA: (**e**) on the absolute coordinates $\sigma-e$; (**f**) on the relative coordinates $\overline{\sigma}-\overline{e}$ ($1 - P = 0.42\%$; $2 - P = 99.58\%$).

## 4. Statistical Assessment of Low-Cycle Fatigue Curves

Probability values enabling the calculation of the theoretical low-cycle fatigue curves and their assessment from the probability perspective were determined for the mechanical properties already investigated statistically (Table 8).

**Table 8.** Probability values of mechanical properties.

| Mechanical Property | Material | Probability, % | | | | | | |
|---|---|---|---|---|---|---|---|---|
| | | 1 | 10 | 30 | 50 | 70 | 90 | 99 |
| $\sigma_{0.2}$, MPa | 15Cr2MoVa | 300 | 340 | 370 | 400 | 430 | 475 | 535 |
| | C45 | 220 | 265 | 300 | 340 | 360 | 420 | 500 |
| | D16T1 | 260 | 300 | 320 | 350 | 370 | 405 | 460 |
| $\sigma_u$, MPa | 15Cr2MoVa | 500 | 530 | 560 | 580 | 600 | 640 | 680 |
| | C45 | 620 | 700 | 750 | 800 | 850 | 900 | 1020 |
| | D16T1 | 580 | 620 | 650 | 680 | 700 | 750 | 800 |
| $\psi$, % | 15Cr2MoVa | 74 | 76 | 79 | 80 | 82 | 85 | 90 |
| | C45 | 28 | 32 | 37 | 39 | 42 | 47 | 54 |
| | D16T1 | 9.5 | 11.3 | 12.8 | 14.0 | 15.5 | 17.5 | 21.0 |
| $e_{pr}$, % | 15Cr2MoVa | 0090 | 0130 | 0170 | 0200 | 0245 | 0320 | 0475 |
| | C45 | 0140 | 0180 | 0225 | 0260 | 0300 | 0360 | 0480 |
| | D16T1 | 0.46 | 0.52 | 0.56 | 0.60 | 0.64 | 0.70 | 0.78 |

For a reliable statistical assessment of low-cycle fatigue durability properties, the paper investigated the distribution patterns and statistical parameters of the mechanical and low-cycle (strain and strength) properties of the materials with contrasting cyclic properties (hardening—aluminium alloy D16T1; softening—steel 15Cr2MoVa; and stable—steel C45).

The Coffin–Manson equation used in the strength calculations defines the dependence of durability under loading with controlled strain ($e_k$ = constant) on the cyclic plastic strain $e_0$ [35,36]:

$$e_0 N_c{}^m = \frac{1}{2} \ln \frac{1}{1-\psi} = C_\psi. \tag{10}$$

The modified Coffin–Manson equation was used in the present study:

$$e_0 N_c{}^{\alpha_{1p}} = C_{1p}. \tag{11}$$

In the equation $\alpha_{1p} < m$ and $C_{1p} < \psi$. Constants $\alpha_{1p}$ and $C_{1p}$ may be determined using the mechanical properties of materials:

$$\alpha_{1p} < 0.17 + 0.55\psi \frac{\sigma_{0.2}}{\sigma_u}, \ C_{1p} = 0.75\alpha_{1p} \frac{100}{\psi_k}. \tag{12}$$

Manson–Langer power equations define cyclic resistance to failure and the dependence of durability between the elastoplastic strain $e_0 = e_p + e_y$ and number of cycles $N$. Under low-cycle loading with controlled strain:

$$e_0 = \frac{1}{4N^m} \ln \frac{100}{100 - \psi_k} + 0.4\frac{\sigma_u}{E}. \tag{13}$$

The study also employed the durability dependence presented in the design rules for the nuclear power industry, PNAE (Regularities and Norms in Nuclear Power Engineering) [37]:

$$e_0 = \frac{0.5 \ln[1/(1-\psi)]}{(4N^m)^{0.5}} + \frac{\sigma_u}{E(4N)^{0.05}}. \tag{14}$$

Using Equations (11), (13), and (14), low-cycle fatigue probability 1%, 10%, 30%, 50%, 70%, 90%, and 99% low-cycle durability curves were designed for steel 15Cr2MoVa, steel C45, and aluminium alloy D16T1 on the relative coordinates $\log\overline{e_0}$–$\log N_c$ (Figures 9 and 10). The relative values $\overline{e_0}$ of plastic strain were obtained by dividing the absolute strain values by the proportional limit strain $e_{pr}$ of the materials.

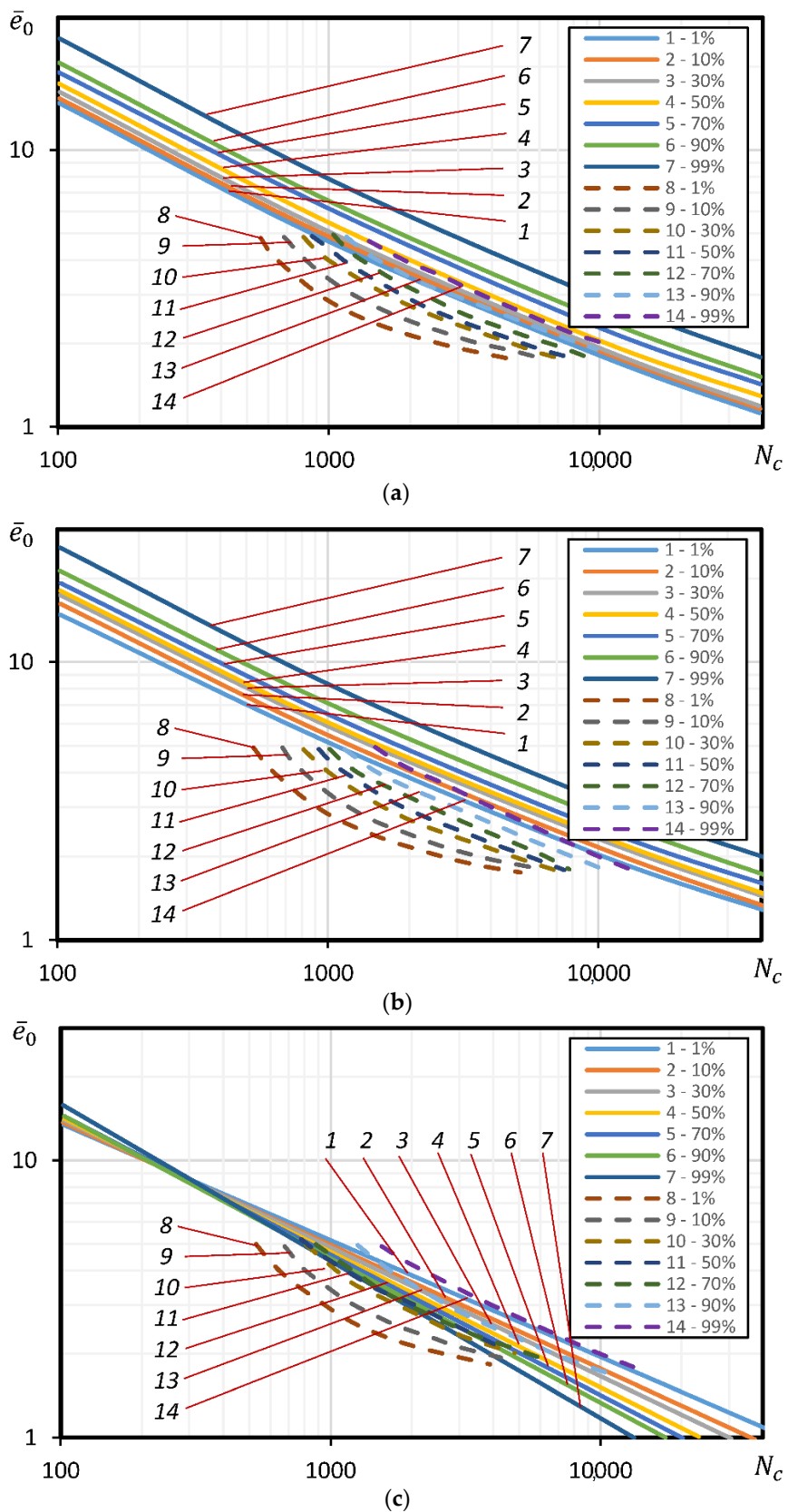

**Figure 9.** Experimental (dashed lines) and theoretical (straight lines) curves for 15Cr2MoVa steel under loading with controlled strain (1–7 = analytical probability, 1–99%; 8–14 = experimental probability, 1–99%): (**a**) according to Coffin dependency (Equation (11)); (**b**) according to Manson–Langer dependency (Equation (13)); (**c**) according to PNAE rules (Equation (14)).

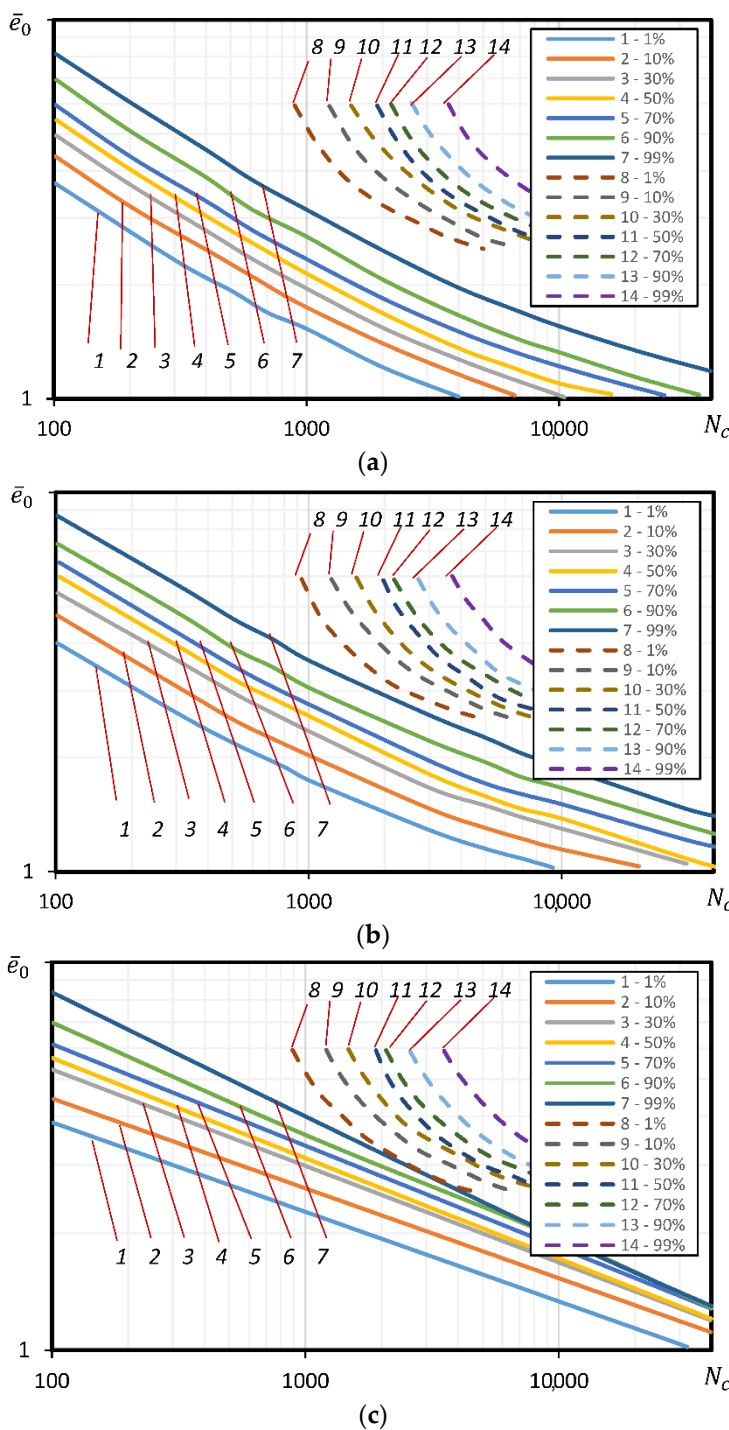

**Figure 10.** Comparison of the experimental (dashed lines) and theoretical (straight lines) curves for the C45 steel under loading with controlled strain (1–7 = analytical probability, 1–99%; 8–10 = experimental probability, 1–99%): (**a**) according to the Coffin dependency (Equation (11)); (**b**) according to the Manson–Langer dependency (Equation (13)); (**c**) according to PNAE rules (Equation (14)).

The comparison of experimental and theoretical curves for steel 15Cr2MoVa presented in Figure 9 showed that the curve slope angle was similar for all cases. However, the experimental curves were slightly lower than the theoretical ones. The theoretically calculated curves fell within the experimental curve zone (Figure 9a); however, they were hardly comparable due to the specifics of the calculation of probability constants $\alpha_{1p}$ and $C_{1p}$ with a small number of cycles. With the number of cycles $N > 400$, the experimental 99% probability curve corresponded to the 50% theoretical curve (Figure 9a). The theoretical

curves calculated under the PNAE rules (Figure 9c) fell between the experimental curves. In all the calculations, the resulting arrangement was the reverse. The 99% experimental curve corresponded to the 1% theoretical curve, etc. It could be assumed that this resulted from the dependence of constant $\alpha_{1p}$ on the relative percent area reduction $\psi$. According to Table 8, the ratio of proportional limit strain $e_{pr}$ 99% to 1% values was 5.3:1 and for the relative percent area reduction it was 1.2:1. Moreover, the proportional limit strain $e_{pr}$ values were sensitive to variations in chemical composition, thermal processing technologies, surface hardening, loading conditions, and other factors of the material.

The comparison of the experimental and theoretical curves of steel C45 under loading with controlled strain ($e_k$ = constant) in Figure 10a suggests that, in all cases, the resulting curve slope was similar. The theoretically calculated curves were lower than the experimental ones; however, in this case, the probability arrangement of the curves was not the reverse. The 99% to 1% durability curve ratio is 7.1:1 at the relative strain amplitude $\overline{e_0}$ = 4% calculated by Equation (11), 7.6:1 according to Equation (12), and 10.3:1 according to Equation (12). In this case, with the relative strain amplitude $\overline{e_0}$ = 2%, the ratios of the durability curves were 8.4:1, 11.7:1, and 5.3:1. Figure 10c suggests that the calculation under the PNAE rules, Equation (15), had the best correspondence with the experimental results.

The results for the D16T1 aluminium alloy under loading with controlled strain ($e_k$ = constant) are presented in Figure 11. A comparison of the 99% and 1% probability curves in Figure 11a showed the clear dependence of the low-cycle durability on the strain level. The conducted analysis suggested that the 99% to 1% durability curve ratio was 37:1, where the strain amplitude $e_0$ = 0.3%, and 24:1 where the strain amplitude $e_0$ = 0.18%. The slope of the theoretical curves increased with the increase in the low-cycle failure probability. This could be related to the percent area reduction $\psi$ distribution (Table 3). The $\overline{e_0}$ distribution band became narrower when relative coordinates were used. The 99% to 1% durability curve ratio was 3.3:1 when the strain amplitude $\overline{e_0}$ = 4 and 2.7:1 when the strain amplitude $\overline{e_0}$ = 3. The slope angles were smaller in the relative coordinate curves. Figure 11b presents a comparison of the experimental and theoretical curves. The experimental and theoretical results differed considerably, with the theoretical curves being in the elasticity zone.

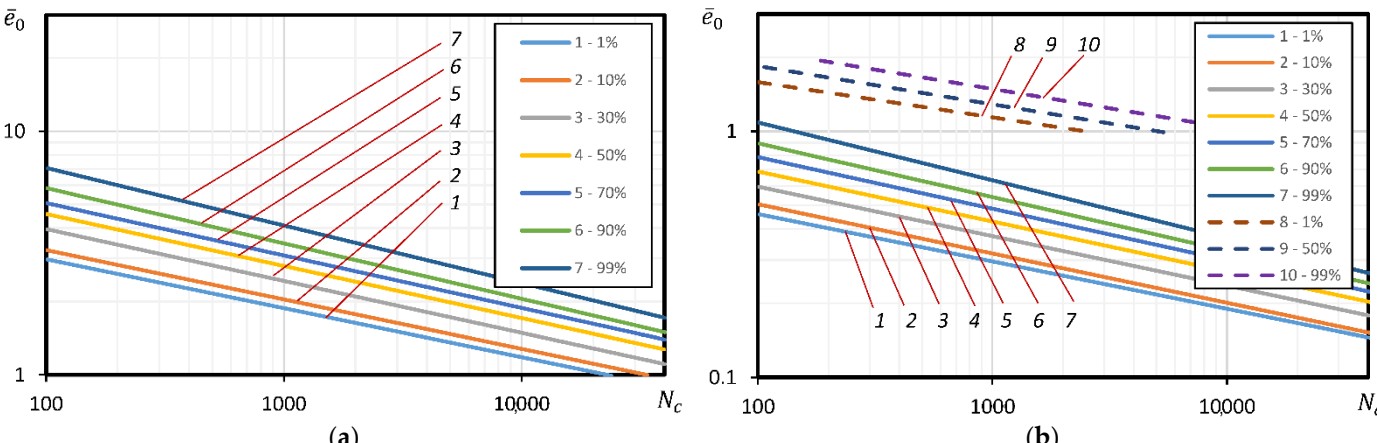

**Figure 11.** Comparison of the theoretical and experimental curves for the D16T1 aluminium alloy under loading with controlled strain (1–7 = analytical probability, 1–99%, straight lines; 8–10 = experimental probability, 1–99%, dashed lines): (**a**) absolute coordinates; (**b**) relative coordinates of the theoretical curve according to the Coffin dependency (Equation (11)).

## 5. Conclusions

After the investigation of the main mechanical characteristics of the materials, histograms were generated and the statistical parameters were evaluated. The statistical analysis of the main mechanical characteristics showed that the sample size does not have

a significant effect on the distribution of the main mechanical characteristics. As a result of the statistical analysis of mechanical properties ($\sigma_{pr}$, $\sigma_{0.2}$, $\sigma_u$, $S_k$, $\psi$, and $\psi_u$) of the materials, coefficient values with minimal variation were found to have been obtained under log-normal distribution. In the case of under loading with controlled strain, the coefficient of variation did not depend on the loading level. An increase in the sample size with loading under controlled strain and controlled stress led to a better correspondence of the statistical series with a normal distribution.

We recommend replacing the mean value of the mechanical property with the lower endpoint of the confidence interval and the standard deviation of the mechanical property with the upper endpoint of the confidence interval.

The comparison of standard, normalised, and experimental mechanical properties showed that the application of the standard properties presented in the specifications to the low-cycle fatigue calculations may lead to significant deviations from the actual results.

The investigations showed the lower distribution of the durability parameters compared to the distribution of parameters of the strain diagrams.

A comparison of the low-cycle fatigue curves demonstrated that the durability curves designed for certain materials using the analytical expressions were not accurate. According to the analysis of relative values of the experimental probability low-cycle fatigue curves calculated by Equations (12), (14), and (15), considerable error may result from the use of analytical dependences for designing the curves. Calculations of critical structures require the use of experimental values corresponding to 50% probability. The paper found a 40–60% correspondence of the mean values resulting from standard investigations of mechanical properties using 3–5 specimens or the investigations of low-cycle fatigue properties and durability using 15–20 specimens to the probability values.

**Author Contributions:** Conceptualization, Ž.B. and V.L.; methodology, Ž.B. and V.L.; software, V.L. and R.Č.; validation, Ž.B. and V.L.; formal analysis, Ž.B. and V.L.; investigation, Ž.B., V.L. and A.V.; resources, A.V.; data curation, Ž.B., V.L. and R.Č.; writing—original draft preparation, Ž.B., V.L. and R.Č.; writing—review and editing, V.L.; visualization, V.L. and R.Č.; supervision, Ž.B. and V.L.; project administration, A.V. All authors have read and agreed to the published version of the manuscript.

**Funding:** This research received no external funding.

**Institutional Review Board Statement:** Not applicable.

**Informed Consent Statement:** Not applicable.

**Data Availability Statement:** Not applicable.

**Conflicts of Interest:** The authors declare no conflict of interest.

## Nomenclature

| | |
|---|---|
| $b$ | tabular parameter |
| $c$ | tabular parameter |
| $c_0$ | the normalised value of the random measure |
| $D$ | dispersion |
| $e$ | monotonous strain (%) |
| $e_0$ | cyclic elastoplastic strain (%) |
| $e_k$ | cyclic strain of $k$ semicycle (%) |
| $e_p$ | cyclic plastic strain (%) |
| $e_{pr}$ | proportional limit strain (%) |
| $e_y$ | cyclic elastic strain (%) |
| $\bar{e}$ | normalised cyclic strain (%) |
| $\bar{e}_0$ | normalised to proportional limit elastoplastic strain (%) |
| $K$ | the ratio of maximum and minimum of the mechanical properties |
| $k_b$ | tabular parameter |
| $k_1$ | statistical data density quantile |
| $m$ | constant |

| | |
|---|---|
| $m^3$ | the third central moment of distribution |
| $m_i$ | number of results in the *i*-th interval |
| $n$ | the total number of the results obtained for particular characteristics |
| $n_a$ | minimum statistical quantity of samples |
| $n_{\text{int}}$ | quantity of statistical intervals |
| $N_c$ | number of load cycles until crack initiation |
| $N_f$ | number of cycles till the cracks propagated to complete fracture |
| $P$ | probability |
| $P_i$ | the density of statistical data |
| $s$ | standard deviation |
| $S$ | skewness |
| $S_k$ | cyclic stress of *k* semicycle (MPa) |
| $t_{\gamma 1}$ | Student's *t*-distribution |
| $t_{1-\gamma/2}$ | quantile of normal distribution |
| $x_i$ | random variable |
| $x_{int}$ | width of the bins (statistical intervals) |
| $x_{max}$ | maximum values of material mechanical properties in the bins (statistical intervals) |
| $x_{min}$ | minimum values of material mechanical properties in the bins (statistical intervals) |
| $x_p^L$ | the lower endpoint of the confidence intervals |
| $x_p^U$ | the upper endpoint of the confidence intervals |
| $\bar{x}$ | sample mean |
| $V$ | coefficient of variation |
| $q_b$ | tabular parameter |

Greek symbols

| | |
|---|---|
| $\gamma$ | reliability of normal distribution |
| $\Delta_a$ | the error of determination of the mean value of the random variable |
| $\psi$ | percent area reduction (%) |
| $\psi_u$ | percent area reduction at failure (%) |
| $\mu$ | arithmetic mean |
| $\sigma$ | monotonic stress (MPa) |
| $\sigma_{0.2}$ | elastic limit or yield strength (MPa), the stress at which 0.2% plastic strain occurs |
| $\sigma_{pr}$ | proportional limit stress (MPa) |
| $\sigma_u$ | ultimate tensile stress (MPa) |
| $\bar{\sigma}$ | normalised to proportional limit cyclic stress (MPa) |

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
