# Peer review of "Probability Assessment of the Mechanical and Low-Cycle Properties of Structural Steels and Aluminium"

_metals, doi:10.3390/met11060918_

Round 1

Reviewer 1 Report

The article is presented with great care, and deserves recognition. All the illustrations are well-detailed, the content does not raise much doubt in understanding the research topic. The article deserves to be published, after the authors address some of the following comments that need improvement.

  1. I think at the end of the phrase "The strength safety margin of structural elements is, in turn, necessary in the assessment of the reliability of operation of the critical structures." you can cite scientifically relevant references (DOI): 10.1016/j.compstruct.2015.06.058, 10.1016/j.compstruct.2015.11.018, 10.1016/j.compstruct.2020.113303.
  2. On page 4 and 5 two figures are labeled figure 2, please change the "second" figure labeled figure 2 to figure 3.
  3. Figure 12a,b,c should be shown enlarged, preferably on one whole page for better readability. The same applies to the figure 13.
  4. Figure 12a,b,c should be shown enlarged, preferably on one whole page for better readability.

Author Response

The corresponding changes in the manuscript are highlighted in red. 

Point 1: I think at the end of the phrase "The strength safety margin of structural elements is, in turn, necessary in the assessment of the reliability of operation of the critical structures." you can cite scientifically relevant references (DOI):

10.1016/j.compstruct.2015.06.058,

10.1016/j.compstruct.2015.11.018,

10.1016/j.compstruct.2020.113303.

Response 1: References have been added.

Point 2: On page 4 and 5 two figures are labeled figure 2, please change the "second" figure labeled figure 2 to figure 3.

Response 1: Figure label has been changed.

Point 3: Figure 12a,b,c should be shown enlarged, preferably on one whole page for better readability. The same applies to the figure 13.

Response 1: Figures 12 a,b,c and 13 a,b,c have been enlarged.

Reviewer 2 Report

  1. The current study investigates the effect of mechanical properties of stress (proportional limit stress σpr, relative yield strength σ0.2, ultimate tensile stress σu) and strain ((proportional limit strain epr, percent area reduction ψ, and percent area reduction at failure ψu) on the low cycle damage in structural steel and aluminium materials. For this the authors, analyse the influence mechanical properties and distribution pattern in order to better understand the material performance.
  2. The abstract must be improved, it needs to be clearer. Lines 8-16 is very long general info which can be reduced.
  3. Please consider reviewing the abstract and highlight the novelty, major findings and conclusions.
  4. Line 52 to 86 I have never seen such literature review before in mdpi journals. The authors must follow the guidelines of metals journal and make sure to write literature review in proper way. Using the study design, findings and value structure is not acceptable.
  5. The literature review must be expanded, the authors only discuss like 3-4 studies which is not acceptable. The authors must provide comprehensive literature review on past studies similar to this work, report what they did and what were their main findings. Then explain how this study brings new knowledge and difference to the field.
  6. After line 86 the authors must answer the following question: What is the research gap did you find from the previous researchers in your field? Mention it properly. It will improve the strength of the article.
  7. Table 1 and Table 2 are missing references? Where did the authors get these values from?  
  8. Moderate English changes required
  9. Extensive editing of English language and style required
  10. Figure 2 is repeated twice? Where is Figure 3?
  11. These histogram figures from Figure 2-4 can the authors consider keeping one of each and move the remaining to Appendix? They are the same and provide same message. Perhaps best is to keep one of each and refer the remaining to the appendix.
  12. Line 164-165 cant we conclude the same from just looking at the histograms?
  13. Line 178-180 can you support this claim with a references(s) from books or previous studies.
  14. Figure 5 please replace numbers from 1-6 in the figure with the actual meaning instead of showing this in the figure caption.  
  15. Line 191 “durability is very likely” why it is very likely? Please elaborate further about this point
  16. Please consider combining figures 6 and 7 into one figure or move some to appendix, if you show one or two of those six graphs in larger size it will be better for the readers to interpret data easily rather than looking at the current ones which are small and not very easy to read due to their compactness.
  17. Line 279-280 please support this claim with a reference and explain further
  18. Combine figures 9 and 10 and 11 into one figure, units are missing in Y axis in all figures please check if this is the case
  19. Line 359 what does the authors mean by was not reverse, please elaborate further
  20. Line 370 please support this claim with a reference “could be related to the percent area reduction”
  21. The results are merely described and is limited to comparing the experimental observation. The authors are encouraged to include more detailed discussion and critically discuss the observations from this investigation with existing literature.
  22. A lot of the math can be moved to the appendix as well.

Author Response

The corresponding changes in the manuscript are highlighted in blue. Some of the comments might overlap with the other reviewers, and could be highlighted using a different color.

Point 1. The current study investigates the effect of mechanical properties of stress (proportional limit stress σpr, relative yield strength σ0.2, ultimate tensile stress σu) and strain ((proportional limit strain epr, percent area reduction ψ, and percent area reduction at failure ψu) on the low cycle damage in structural steel and aluminium materials. For this the authors, analyse the influence mechanical properties and distribution pattern in order to better understand the material performance.

Point 2. The abstract must be improved, it needs to be clearer. Lines 8-16 is very long general info which can be reduced.

Response 2: General info was removed from abstract.

Point 3. Please consider reviewing the abstract and highlight the novelty, major findings and conclusions.

Response 3: Abstract has been rewritten with major findings and conclusions.

Point 4. Line 52 to 86 I have never seen such literature review before in mdpi journals. The authors must follow the guidelines of metals journal and make sure to write literature review in proper way. Using the study design, findings and value structure is not acceptable.

Response 4: Study design, findings and value structure were removed from literature review.

Point 5. The literature review must be expanded, the authors only discuss like 3-4 studies which is not acceptable. The authors must provide comprehensive literature review on past studies similar to this work, report what they did and what were their main findings. Then explain how this study brings new knowledge and difference to the field.

Response 5: Literature review has been expanded.

Point 6. After line 86 the authors must answer the following question: What is the research gap did you find from the previous researchers in your field? Mention it properly. It will improve the strength of the article.

Response 6: Comment has been inserted to the text.

Point 7. Table 1 and Table 2 are missing references? Where did the authors get these values from?

Response 7: Standards for the chemical composition and mechanical properties of materials have been included in the text.

Point 8. Moderate English changes required

Point 9. Extensive editing of English language and style required

Response 8, 9: English literacy will be increased in the next revisions.

Point 10. Figure 2 is repeated twice? Where is Figure 3?

Response 10: Figure 3 caption has been changed.

Point 11. These histogram figures from Figure 2-4 can the authors consider keeping one of each and move the remaining to Appendix? They are the same and provide same message. Perhaps best is to keep one of each and refer the remaining to the appendix.

Response 11: Comparison of histograms is meaningful with each other. Therefore, the separation will mislead the reader.

Point 12. Line 164-165 cant we conclude the same from just looking at the histograms?

Response 12: An approximate conclusion can be made. To improve the statistical assessment of the mechanical properties, statistical characteristics were calculated.

Point 13. Line 178-180 can you support this claim with a references(s) from books or previous studies.

Response 13: Reference has been added.

Point 14. Figure 5 please replace numbers from 1-6 in the figure with the actual meaning instead of showing this in the figure caption.

Response 14: We have tried to keep the style of all figures uniform, i.e. without letters and symbols. There is no reason to make one figure different.

Point 15. Line 191 “durability is very likely” why it is very likely? Please elaborate further about this point

Response 15: When selecting key mechanical properties provided in the specifications, an error may be made due to the failure to account for a series of random factors that determine the distribution of properties. Application of the probability methods enables increasing life characteristics of structures and substantiating the cycle load values.

Point 16. Please consider combining figures 6 and 7 into one figure or move some to appendix, if you show one or two of those six graphs in larger size it will be better for the readers to interpret data easily rather than looking at the current ones which are small and not very easy to read due to their compactness.

Response 16: Figures have been combined and enlarged.

Point 17. Line 279-280 please support this claim with a reference and explain further

Response 17: Reference has been inserted. The article describes the relationship between mechanical and cyclic characteristics.

Point 18. Combine figures 9 and 10 and 11 into one figure, units are missing in Y axis in all figures please check if this is the case

Response 18: Figures have been combined and enlarged. Added units in Y axis.

Point 19. Line 359 what does the authors mean by was not reverse, please elaborate further

Response 19: Reversed arrangement of the curves, i.e. the lower curve has 99% probability and the upper curve has 1% probability.

Point 20. Line 370 please support this claim with a reference “could be related to the percent area reduction”

Response 20: Reference has been inserted to the text.

Point 21. The results are merely described and is limited to comparing the experimental observation. The authors are encouraged to include more detailed discussion and critically discuss the observations from this investigation with existing literature.

Response 21:  Several additional findings have been added to the abstract and conclusions section. But in general, we would wish to keep the conclusions focused on the most relevant issues related to the paper.

Point 22. A lot of the math can be moved to the appendix as well.

Response 22: Symbols used in the math already moved and explained in section Nomenclature. We consider that several references in the text will mislead the reader.

Reviewer 3 Report

This paper presented a probability assessment of the low-cycle fatigue properties of steel and aluminum allow. However, the paper needs a massive restructure considering the following before rereviewed again:

  1. The literature review is not current. A total of 28 references were used mostly10-40 years old. Only two references were found from the year 2020-2021.
  2. The experimental section is not clear and not reproducible. How a reader will be able to do the same experiment provided in this manuscript?
  3. No standard name of structural steel of C45 is given.
  4.  A mixture of aluminium vs aluminum is used throughout the paper.
  5. Page 3 line 122- does not make any meaning. Is not D16T1 an aluminum alloy?
  6. Why a different number of samples used for different materials?
  7. Equation 1 is not clear. what do you mean by the length of the statistical interval and maximum values of material mechanical properties in a statistical row and so on.. Just impossible to understand.
  8.  Figure 2 is presented twice and no figure 3 is available.
  9.  No dotted line in figure 14. The caption does not match with the figure.

Author Response

The corresponding changes in the manuscript are highlighted in light blue. Note that some of the comments might overlap with the other reviewers, and could be highlighted using a different color.

Point 1: The literature review is not current. A total of 28 references were used mostly10-40 years old. Only two references were found from the year 2020-2021.

Response 1: New articles have been included.

Point 2: The experimental section is not clear and not reproducible. How a reader will be able to do the same experiment provided in this manuscript?

Response 2: The fatigue and tensile experiments were performed according to the methodology specified in the standard GOST 25502 – 79.

Parallel standards:

ISO 6892-1:2016. Metallic Materials—Tensile Testing—Part 1: Method of Test at Room Temperature; International Organization for Standardization (ISO): Geneva, Switzerland, 2016.

ISO-1099. Metallic Materials—Fatigue Testing—Axial Force-Controlled Method; International Organization for Standardization: Geneva, Switzerland, 2006.

After the fatigue tests the fractured specimens were used as the workpiece materials to produce monotonous tensile specimens with the aim of reaching the material properties nearly identical to the properties of the material subjected to cyclic loading.

Point 3: No standard name of structural steel of C45 is given.

Response 3: Standards for the chemical composition and mechanical properties of materials have been included to the text.

Point 4: A Page 3 line 122- does not make any meaning. Is not D16T1 an aluminum alloy?

mixture of aluminium vs aluminum is used throughout the paper.

Response 4: D16T1 is an aluminium alloy. A description has been inserted to the text. The term “aluminium” has been changed throughout the paper.

Point 5: Why a different number of samples used for different materials?

Response 5: It is necessary to have at least 10 samples in order to obtain reliable results according to the statistical test procedure described in the standard.

The aim to use a different number of samples was to determine the effect of the sample size on the statistical results. The study showed that the number of samples does not have a significant effect. A brief description has been inserted in the text.

Point 6: Equation 1 is not clear. what do you mean by the length of the statistical interval and maximum values of material mechanical properties in a statistical row and so on. Just impossible to understand.

Response 6:  Changed descriptions of the histograms variables. Term “statistical row” has been changed to “bin (statistical interval)”. Term “the length of the statistical interval” has been changed to “bin width”. 

Point 7: Figure 2 is presented twice and no figure 3 is available.

Response 7: Figure caption has been changed.

Point 8: No dotted line in figure 14. The caption does not match with the figure.

Response 8: Figure 14 caption has been changed.

Round 2

Reviewer 2 Report

All questions answered. please try to improve the English 

congratulations to the authors

Author Response

The English language has been improved.

Reviewer 3 Report

The authors have improved the quality of the presentation. It can be accepted now. However, Please move the nomenclature at the start of the manuscript, so that the reader can find the meaning of symbols easily.

Author Response

Point 1: Please move the nomenclature at the start of the manuscript, so that the reader can find the meaning of symbols easily.

Response 1: The nomenclature has been moved at the start of the manuscript.